# Position: Bridge the Gaps between Machine Unlearning and AI Regulation

Bill Marino[1,*]      Meghdad Kurmanji[1,*]      Nicholas D. Lane[1]

## Abstract

The "right to be forgotten" and the data privacy laws that encode it have motivated machine unlearning since its earliest days. Now, some argue that an inbound wave of artificial intelligence regulations — like the European Union's Artificial Intelligence Act (AIA) — may offer important new use cases for machine unlearning. However, this position paper argues, this opportunity will only be realized if researchers proactively bridge the (sometimes sizable) gaps between machine unlearning's state of the art and its potential applications to AI regulation. To demonstrate this point, we use the AIA as our primary case study. Specifically, we deliver a "state of the union" as regards machine unlearning's current potential (or, in many cases, lack thereof) for aiding compliance with various provisions of the AIA. This starts with a precise cataloging of the potential applications of machine unlearning to AIA compliance. For each, we flag the technical gaps that exist between the potential application and the state of the art of machine unlearning. Finally, we end with a call to action: for machine learning researchers to solve the open technical questions that could unlock machine unlearning's potential to assist compliance with the AIA — and other AI regulations like it.

## 1 Introduction

Since its inception, Machine Unlearning (MU) has been motivated by the so-called "right to be forgotten" (RTBF) [12], which is encoded in data privacy laws like the European Union's General Data Protection Regulation (GDPR) [35, Art. 17]. Now, a new wave of AI regulations — including but not limited to the European Union's Artificial Intelligence Act (AIA) [37] — are working their way through the legislative process [8, 6] or have graduated it and gone into effect [24, 7]. As these AI regulations take shape, researchers have begun to explore whether MU can play a role in supporting compliance with them too [79, 91, 46, 94, 89, 60, 64]. However, recent works call into question whether this motivation really holds water [26].

**In this position paper, we argue that MU's potential to assist compliance with AI regulation will only be realized if researchers close the technical gaps between MU's state of the art and this prospective new application.**

We use the AIA as an example to support our argument. This starts with a thorough cataloging of the AIA requirements that MU can hypothetically assist compliance with. For each of these potential use cases, we then scrutinize, from a technical perspective, whether the state of the art (SOTA) of MU can really support the hypothesized application, with a special eye towards auditability (which is especially relevant in the regulation context). In many cases, we identify considerable gaps between the two. Finally, we conclude with a

---

*Equal contribution. [1] University of Cambridge. Correspondence to: `wlm27@cam.ac.uk`.

39th Conference on Neural Information Processing Systems (NeurIPS 2025) Position Paper Track.

pointed call for the AI research community to take action and fill these gaps in order to help make MU a more viable tool for assisting AI regulation compliance.

## 2    Machine Unlearning

To set the stage for our analysis, in this section, we define and provide an overview of MU and its key concepts:

### 2.1    Formal Definition of Unlearning

Let $M = A(D)$ denote a model trained on dataset $D$ using algorithm $A$. An **unlearning query** specifies a **forget-set** $D_f \subset D$, with the **retain-set** defined as $D_r = D \setminus D_f$. The goal of an unlearning algorithm $U$ is to remove the influence of $D_f$ from $M$, yielding an unlearned model $M_u = U(M; D_f, D_r)$. Depending on the approach, $U$ may not require access to $D_r$ [136].

**Definition 2.1.** Following [47], $U$ is an $(\epsilon, \delta)$-**unlearner** if the distribution of $U(M; D_f, D_r)$ is $(\epsilon, \delta)$-close to that of $A(D_r)$. Specifically, two distributions $\mu$ and $\nu$ are $(\epsilon, \delta)$-close if for all measurable events $B$:  $\mu(B) \leq e^\epsilon \nu(B) + \delta$   and   $\nu(B) \leq e^\epsilon \mu(B) + \delta$.

This definition provides a natural taxonomy for MU algorithms. When $\epsilon = \delta = 0$, $U$ is termed **exact unlearning**; otherwise, it is referred to as **approximate unlearning**.

### 2.2    Informal Definitions

While Def. 2.1 provides a rigorous formulation of MU, researchers commonly use informal interpretations, typically phrased as `removing x from M`. However, deriving informal definitions directly from Def. 2.1 can be challenging, as the entity to remove may not be explicitly identifiable. For example, in generative models, $x$ often corresponds to a fact or concept without explicit representation in $M$ or $D$.

Additionally, MU is broadly applied to various methods, but overly general definitions introduce unnecessary complexity, potentially obstructing clear scientific discourse. Therefore, we restrict MU in this paper to ML techniques that explicitly modify the model's parameter-set (e.g., deletion and retraining, fine-tuning, parameter addition or removal). This scoped definition allows MU to remain a practical tool for applications such as safeguarding and alignment, while methods like guardrailing (or "output suppression" as per Cooper et al. [26]) remain distinct, meriting separate evaluation in regulatory and other contexts.

### 2.3    Evaluation metrics

While Def. 2.1 is widely accepted in the MU community, it presents several challenges in practical settings. First, some works question whether this definition is necessary or sufficient to achieve true MU [111]. Second, in large-scale applications, it is computationally infeasible to directly measure the closeness between the distributions $A(D_r)$ and $U(M; D_f, D_r)$. As a result, researchers often resort to alternative proxies to verify MU. These proxies include performance metrics (e.g., classification accuracy [50] or generative performance using metrics like ROUGE for large language models [88]) and privacy attacks, such as membership inference attacks [61, 113].

### 2.4    Trade-offs and risks

MU algorithms strive balance three key objectives: **Model Utility**, **Forgetting Quality**, and **Efficiency**. In certain privacy-centric applications, forgetting can be synonymous with achieving privacy [83]. Hyperparameters and regularizers impact these trade-offs. For example, in MU via `Fine-tune`, the number of steps and learning rate dictate the balance between forgetting quality and efficiency [127]. Similarly, in `Gradient Ascent`, the number of steps determines the trade-off between effective MU and preserving model's utility [74].

Additionally, forgetting may sometimes conflict with privacy due to two phenomena. First, unlearning specific data points can inadvertently expose information about others in the

retained set due to the "onion effect" of privacy [14]. Second, over-forgetting [74] a data point may reveal its membership in the original training set—a phenomenon known as the "Streisand Effect" [50]. Addressing these challenges requires careful calibration of MU methods to ensure a delicate equilibrium across these competing objectives.

Beyond these trade-offs, MU introduces risks associated with *untrusted parties* [77] and *malicious unlearning* [100]. Malicious entities could exploit MU to make fake deletion queries, or introduce computation overhead to systems [64].

# 3 The EU's Artificial Intelligence Act

The AIA sets forth requirements for AI systems and models placed on the market or put into service in the EU [37, Art. 1]. These requirements largely target two categories of AI: AI systems and general-purpose AI ("GPAI") models. Here, we define these categories and, for each, review some the AIA requirements that relate to the discussion at hand.

## 3.1 AI Systems

The AIA broadly defines AI systems to include any "machine-based system that is designed to operate with varying levels of autonomy and that may exhibit adaptiveness after deployment, and that, for explicit or implicit objectives, infers, from the input it receives, how to generate outputs such as predictions, content, recommendations, or decisions that can influence physical or virtual environments" [37, Art. 3.1]. An example of a system that might meet this criteria is ChatGPT [41].

In laying out its rules for these AI systems, the AIA relies on a "risk-based" approach [87], by which an AI system's perceived degree of risk determines the exact rules that apply to it. Here, the strictest requirements — and the ones most relevant to our discussion — are reserved for those AI systems deemed to be *high-risk* [37, Art. 6]. Such high-risk AI ("HRAI") systems are subject to a bevy of requirements [37, Chap. III]. Among them, the following are the most relevant to our position:

**Risk management**: HRAI systems must implement risk management systems that identify known and reasonably foreseeable risks that the system may pose to health and safety or to fundamental rights [37, 71, Art. 9.2.a]. Here, risks to *health and safety* includes risks to mental and social well-being as well as physical safety. [2, 33]. Meanwhile, risks to *fundamental rights* includes, among other things, the right to non-discrimination [34], including from biased results [3]. Importantly, wherever these risks are identified, they should be "reasonably mitigated or eliminated through the development or design" of the AI system [37, Art. 9.2-3].

**Accuracy and cybersecurity**: HRAI systems must be designed and developed so as to achieve an "appropriate level" of accuracy and cybersecurity [37, Art. 15.1]. In its Recitals, the AIA stresses that these appropriate levels are a function of the system's intended purpose as well as the SOTA [37, Rec. 74]. When it comes to cybersecurity, the AIA specifically requires that HRAI systems be "resilient against attempts by unauthorised third parties to alter their use, outputs or performance by exploiting system vulnerabilities" [37, Art. 15.5] and take technical measures to "prevent, detect, respond to, resolve and control for ... data poisoning" as well as "confidentiality attacks" [37, Art. 15.5].

## 3.2 GPAI models

In contrast to an AI system, a GPAI model is defined as any AI model that is "trained with a large amount of data using self-supervision at scale, that displays significant generality and is capable of competently performing a wide range of distinct tasks regardless of the way the model is placed on the market and that can be integrated into a variety of downstream systems or applications, except AI models that are used for research, development or prototyping activities before they are placed on the market" [37, Art. 3.63]. Some see this as being synonymous with "foundation model" [1]. An example of a GPAI model that might meet this criteria is GPT 3.5, the model powering ChatGPT [41].

In laying out its requirements for GPAI models, the AIA again uses a risk-based approach, with the strictest requirements reserved for GPAI models deemed to carry *systemic risk* [37, Art. 55]. This is defined as the risk of "having a significant impact on the [EU] market due to [its] reach, or due to actual or reasonably foreseeable negative effects on public health, safety, public security, fundamental rights, or the society as a whole, that can be propagated at scale across the value chain" [37, Art. 2.65; Annex III]. This status can be established through proxies, including performance on benchmarks and the amount of compute used during training [37, Art. 51]. While the AIA itself does not further elaborate on what constitutes systemic risk, a companion piece to the AIA posits that it covers risks related to: (1) cyber offense; (2) chemical, biological, radiological, and nuclear (CBRN); (3) loss of control; (4) automated use of models for AI research and development; (5) persuasion and manipulation; and (6) large-scale discrimination [38].

Among the AIA's requirements for GPAI models that display systemic risk — and those that don't — the following are the most relevant to our analysis:

**Copyright**: All GPAI model providers must "put in place a policy to comply with Union law on copyright and related rights" [37, Art. 53.c]. Among other things, this policy must respect rightsholders' requests, per [36, Art. 4.3], to opt out of text and data mining (TDM) on their copyrighted works [37, Rec. 105, Art. 53.c].

**Systemic risk**: GPAI models that display systemic risk must "mitigate" it [37, Art. 55.a-b].

**Cybersecurity**: GPAI models with systemic risk are additionally required to "ensure an adequate level of cybersecurity" [37, Art. 55(d)].

## 4  MU for AIA compliance: a catalog

This Section comprehensively catalogs the potential applications of MU to assist AIA compliance. For each, we analyze the SOTA and its ability to support the potential application, then identify any open questions the research community must resolve in order to bridge the gap between the (including as regards verification, which can be especially important in the regulatory context — for regulators or any other auditors). In sum, we find that the potential applications of MU to assist AIA compliance ultimately roll up into just six separate applications (Fig. 1):

- **Accuracy:** Improve accuracy per EU [37, Arts. 9, 15];
- **Bias:** Mitigate bias per EU [37, Arts. 9, 55];
- **Privacy Attack:** Mitigate confidentiality attacks per EU [37, Arts. 9, 15, 55];
- **Data Poisoning:** Mitigate data poisoning per EU [37, Art. 15];
- **GenAI risk:** Mitigate other risks of generative outputs per EU [37, Arts. 9, 55];
- **Copyright:** Aid compliance with copyright laws, per EU [37, Art. 53].

We study six applications of MU for AIA compliance; in practice, these are often overlapping or interdependent rather than disjoint. For example, in terms of overlap, one unlearning algorithm such as retraining can be used to address multiple use cases: e.g., accuracy, biases, and copyright. In terms of interdependencies, unlearning mislabeled or stale samples to raise overall accuracy (Sec. 4.1) can shift subgroup error profiles, impacting fairness (Sec. 4.2). Differently, debiasing forget-sets may alter susceptibility to membership inference (Sec. 4.3). Likewise, removing backdoored data (Sec. 4.4) often trades off with clean accuracy (Sec. 4.1).

### 4.1  Accuracy

Two AIA provisions may compel HRAI systems providers towards higher levels of accuracy. First, HRAI systems must achieve a level of accuracy appropriate to their intended use and the SOTA [37, Art. 15.1; Rec. 74]. Second, HRAI systems' risk management practices must include measures to mitigate or eliminate risks to health and safety [37, Art. 9], which, in some domains like medicine, could potentially stem from low accuracy [69, 68, 62]. In either

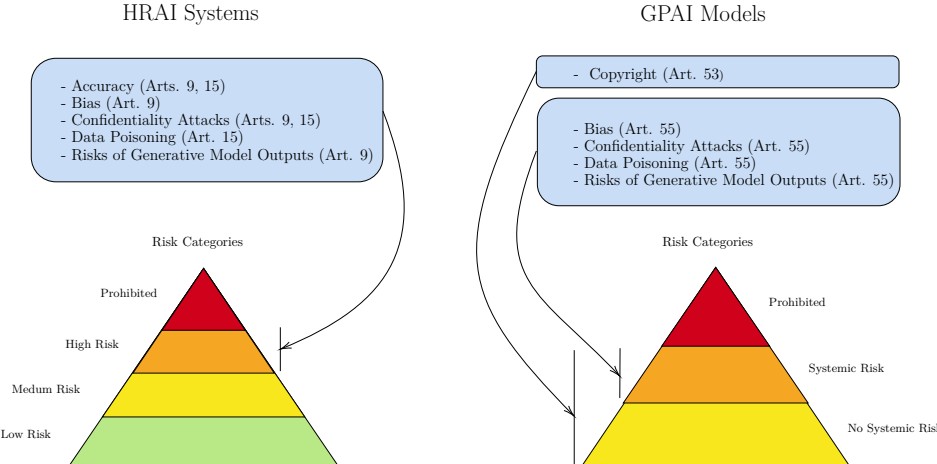

Figure 1: AIA Uses Cases for Machine Unlearning.

case, MU can hypothetically boost accuracy by removing the effect of problematic (e.g., mislabeled) data from the model [110], thus assisting compliance.

This accuracy use case should not require privacy guarantees on the unlearned data [48], because the goal is strictly to boost accuracy to the level deemed appropriate [37, Art. 15] or until the overall residual risk to health and safety posed by the inaccuracy is judged to be "acceptable" [37, Art. 9]. In measuring that, AI providers will presumably account for any inadvertent, counteracting degradation in accuracy caused by the MU itself [76, 11, 125].

**Current SOTA** MU theoretically offer paths towards improving model accuracy by forgetting mislabeled [48, 110, 13, 16, 56], out-of-date and outlier training data points [73, 126, 125, 75], or, potentially, removing noise from medical data [99, 30]. The largest hurdle for this use case might be identifying all of the data points that are leading to inaccuracy (e.g., the mislabeled examples), which can be difficult [49]. It may be good enough to identify only a subset of these examples — so long as accuracy is boosted to levels deemed "appropriate" in light of the intended purpose as well as the SOTA [37, Art. 15.1; Rec. 74]. MU based on subset forget sets have shown success in boosting accuracy; however, other studies have suggested that you need all of polluted data, not just some of this, or it might backfire [49]. It is also important to note that verifying unlearning success is application dependent — and that approximate unlearning should not be expected to yield higher accuracy than exact retraining without the low-quality data.

> **Key Points:** (i) Multiple AIA requirements may benefit from MU. (ii) Theoretical guarantees may not be needed. (iii) Evaluation measure is application-dependent.

> **Open Problems:** (i) Lack of reliable methods for identifying problematic data to unlearn. (ii) Lack of controllability over trade-offs.

### 4.2 Bias

Providers of both HRAI systems and GPAI models with systemic risk must mitigate certain types of model bias. The former must take measures to mitigate or eliminate risks to fundamental rights, which includes the right to non-discrimination [37, Art. 9]. The latter must take steps to mitigate their models' systemic risk [37, Art. 55], which includes the risk of large-scale discrimination [38]. Bias can occur because unrepresentative or incomplete data prevent the model from perform fairly on different groups or, in the case of generative models, cause it to produce stereotyped or otherwise discriminatory outputs [42]. In all cases, MU can ostensibly help forget the data points or training data patterns causing the bias [98, 75, 104, 72, 16, 138]. An important limiting factor on this use case is that training data that is not there to begin with cannot be forgotten; if the bias is due to a data *deficit*, MU

will not help. Because the goal here is to reduce or eradicate bias, success should ultimately be measured and verified using traditional bias metrics like the difference in performance on various subgroups [27] or, in the case of generative models, the propensity for biased outputs as measured with benchmarks [96].

**Current SOTA** Model debiasing has a longer history than MU [130]. Recently, both exact [58]) and approximate [20, 95, 58]) MU solutions have been offered to mitigate model biases. In the debiasing literature, solutions include *pre-processing*, *in-processing*, and *post-processing* methods [90]. MU, can mainly be considered as a post-processing method. However, it is difficult to draw a separating line between debiasing and MU methods. MU works usually re-use some of the evaluation metrics in the debiasing literature, however, how to evaluate bias is, generally, considered an "open problem" [102]. In order to preserve accuracy (also required under the AIA) by not forgetting data points holistically, [124] use MU to forget only those features that lead to bias.

> **Key Points:** (i) MU may aid compliance with multiple bias-related AIA requirements. (ii) MU is only subtractive and never additive, limiting its application to this use case. (iii) De-biasing solutions are not limited to MU.

> **Open Problems** (i) Lack of methods for identifying bias counterfactuals. (i) Lack of controllability over trade-offs. (iii) Difficulty of guaranteeing full unlearning of biases, due to generalization.

## 4.3   Confidentiality attacks

The AIA requires providers of both HRAI systems and GPAI models with systemic risk to resolve and control for confidentiality attacks. Providers of HRAI systems must ensure their systems achieve an "appropriate level" of cybersecurity given the intended use and the SOTA, including by taking technical measures to prevent, detect, respond to, resolve and control for confidentiality attacks [37, Art. 15.5; Rec. 74]. Meanwhile, providers of GPAI models with systemic risk must ensure those models reflect "an adequate level of cybersecurity" [37, Art. 55.d], which presumably also includes defending against confidentiality attacks. While the AIA does not define confidentiality attacks, we take them to include any attacks, including data reconstruction and membership inference attacks, that cause a model to reveal confidential details about its training such as data points or membership [115, 23]. This may include confidential training data memorized by generative models [26, 54, 138]. Where such attacks occur — or where there is reason to think they might — MU can ostensibly help defend against them by forgetting the confidential information vulnerable to attack [64, 75, 14, 18, 126, 102, 5]. For this use case, the measure of success (i.e., verification) should be whether confidentiality attacks succeed in the wake of the MU [53], though use case-specific metrics have been developed [88]. When it comes to this use case, there are, importantly, other viable options for protecting against confidentiality attacks, including training with differential privacy (DP) [117, 70].

**Current SOTA.** Multiple MU techniques have been proposed to mitigate confidentiality attacks (or the related problem of inadvertent model leakage of personal data) [28, 4, 85, 17, 119, 12, 11, 10]. As is, applying MU to this use case can carry sizable trade-offs. For example, unlearning some data points for the sake of protecting them from recovery by attackers can jeopardize the privacy of other data points that neighbor the unlearned ones [14] or even increase the risk of membership inference attacks that recover the unlearned data points [19, 5, 74]. Differently, approximate unlearning, when used to delete particular data points, can carry a bias trade-off [131, 95] and an accuracy trade-off that rises as more data is forgotten [52, 88]; either trade-off can potentially undermine AIA compliance. It is also important to note that current MU methods usually fail on new emergent attacks that are devised with new assumptions [135, 66].

> **Key Points** (i) MU may aid compliance with multiple confidentiality attack-related AIA requirements. (ii) Due to attack diversity, success should be measured on case-by-case basis. (iii) DP is a strong alternative to MU for this use case.

**Open Problems** (i) Difficulty of providing formal guarantees of attack susceptibility. (ii) Difficulty of applying MU to new, emergent attacks. (iii) Identifying, localizing, and measuring memorization of confidential data.

## 4.4 Data poisoning

In data poisoning, specially-crafted data points are injected into a training set to alter (e.g., degrade or bias) model behavior to the attacker's benefit [9]. Backdoor attacks are a type of data poisoning where the injected data points create "triggers" the attacker can exploit during inference [80]. The AIA obligates the providers of both HRAI systems and GPAI model with systemic risk to address such attacks. HRAI system providers must ensure their systems achieve an "appropriate level" of cybersecurity, including via technical measures to "prevent, detect, respond to, resolve and control for" data poisoning attacks [37, Art. 15.5]. Providers of GPAI models with systemic risk, meanwhile, must "ensure an adequate level of cybersecurity" in their models [37, Art. 55.d], which presumably also includes defenses against data poisoning. Where it is known that data poisoning has (or could) occur, MU may help remove the effects of the poisoned data points on the model and, thus, help satisfy these requirements [125, 84, 12, 13, 107]. When it comes to measuring and verifying success for this use case, because the "primary goal is to unlearn the adverse effect due to the manipulated data," the ideal benchmark would seem to be whether those adverse effects — be they vulnerability to backdoor triggers, bias, or lower accuracy — are eliminated or reduced [49]. For example, Goel et al. [49] measure MU efficacy based on whether proper accuracy on backdoor triggers is restored.

**Current SOTA** Though some work has demonstrated MU can succeed for this use case [120, 105] other works question the effectiveness of using MU to address data poisoning or backdoor attacks specifically [55, 112, 97, 126]). As always, identifying the full forget set (here, the poisoned samples) remains challenging [49]. Some methods, moreover, can have a significant accuracy trade-off on this use case [97]. Such trade-offs can be particularly difficult as poisoned data overlaps with the clean data and, in most cases, they are even visually indistinguishable from each other.

**Key Points** (i) MU may aid compliance with several data poisoning-related AIA requirements. (ii) A proper benchmark should measure the elimination of adverse effects.

**Open Problems** (i) Finding contaminated data at scale is challenging. (ii) Unlearning the backdoor pattern without hurting unaffected data is challenging. (iii) Current MU methods mostly fail on data poisoning use case.

## 4.5 Other risks of generative outputs

Generative outputs may pose risks to health, safety, and human rights or pose systemic risk that providers of HRAI systems and GPAI models, respectively, must mitigate. For example, HRAI systems' risk management systems must strive to mitigate or eliminate risks the system poses to health, safety, and fundamental rights [37, Art. 9]. Generative outputs may pose risks to health and safety, e.g., by issuing bad medical advice [122, 59], and may pose risks to the fundamental right of non-discrimination, e.g., by producing stereotyping outputs [92]. For GPAI models with systemic risk, providers of such models must mitigate that risk [37, Art. 55], which could be brought on by generative model outputs that display offensive cyber capabilities, knowledge of CBRN, and more [38, 93]. In all these cases, MU may help mitigate the non-compliant outputs by unlearning the data points or even the concepts in the training set that are causing them [138, 26]. Computationally, it may offer advantages even as compared to other popular alignment techniques like reinforcement learning [128]. Measuring success for this use case should arguably be "context dependent" [128]. That is, the best way to verify the MU's efficacy is to benchmark the exact behavior that we desire to repair [128]. This could utilize existing benchmarks unrelated to MU [5]. Differently, Li et al. [78] propose a benchmark for measuring MU of CBRN knowledge and approaches

that examine the model parameters for remnants of the unlearned concepts have also been proposed [65].

**Current SOTA** Multiple works use MU to curb undesirable generative model outputs [29, 129, 121, 44]. However, the task is difficult, without agreed-upon best practices [26]. Broad concepts like non-discrimination tend to go beyond individual data points, to latent information which is not easily embodied as a discrete forget set [26, 81]. Even if data points that are intrinsically harmful (e.g., the molecular structure of a bioweapon) are removed, models may still assemble dangerous outputs from latent information in the rest of the dataset [26, 114]. Trying to remove that latent knowledge can risk model utility [26]. As a separate but related issue, AI systems in these scenarios may be dual-use, where the appropriateness of outputs depends on downstream context; this, too, makes identifying the forget set difficult and increases the likelihood of a utility trade-off as the model forgets desirable knowledge alongside undesirable knowledge [26, 108, 102]. All of these issues, in turn, make it difficult if not impossible to specify formal guarantees on the MU [81].

> **Key Points** MU may aid compliance with several AIA requirements related to generative outputs.

> **Open Problems** (i) Defining the forget set when what we seek to forget is conceptual. (ii) Difficulty of guaranteeing full unlearning of unwanted behaviors, due to generalization. (iii) Mitigating the forgetting of useful knowledge alongside undesirable knowledge.

## 4.6   Copyright

All GPAI model providers must have a policy for complying with EU copyright law [37, Art. 53.c]. Among other things, this policy must honor the TDM opt-outs of rightsholders [37, Art. 53.c; Rec. 105], which is often a feature of AI training [103, 57]. When it comes to AI and copyright law, a distinction is sometimes made between the "input" (training) phase and the "output" (inference) phase of the AI life cycle [103, 101]. At this point in time, the primary compliance risk during the input phase seems to be that an AI training set could include data points that violate TDM opt-outs. When this happens, we assume that using MU to remove the opt-out data points from the trained model does not cure the violation, since the violation occurred at the moment the opt-out data was used for training. That said, MU may still represent a valuable component of a copyright-compliance policy by helping prevent, at the "output" phase, further violations of copyright law when the opt-out data points — or any other copyright-protected data points in the training set — are reproduced to some degree in model outputs [103]. This is a real risk with generative models, which often memorize training data [25, 15]. When MU is applied to this use case, we may measure success by tracking how likely the model is to generate works that are sufficiently similar to the copyrighted works. For example, we might rely on existing benchmarks that measure the tendency of models to produce copyrighted materials [82, 21]. Differently, Ma et al. [86] produce a benchmark for the success of MU in the copyright context.

**Current SOTA** Wu et al. [123] unlearn copyrighted works from diffusion models. At first glance, exact MU would seem to provide a guarantee that copyrighted works in the training set will not be reproduced in outputs [81]. But the fact is that retraining from scratch without the copyrighted data may not be a bulletproof solution for preventing copyright infringement in outputs because substantially similar representations of copyrighted "expressions" (e.g., images of characters like Spiderman) could still appear in outputs based on how the model generalizes from the latent information extracted from the rest of the training set [26]. For the same reason, approximate unlearning aimed at removing the influence of the copyright data points on the model, on top of being hard to prove [81], also cannot ensure that copyrights are not infringed by outputs. In general, the SOTA of approximate unlearning has been deemed "insufficient" for the copyright use case, which may be why practitioners currently lean towards pre- and post-processing tools like prompting and moderation to bring AI into compliance with these laws [81, 109]. Dou et al. [31] "unlearn" copyrighted materials in LLM pre-training datasets by identifying and removing specific weight updates in the model's parameters that correspond to copyrighted content, evaluating their method by measuring

the similarities between the model's outputs and the original content. The task of measuring whether substantially similar outputs are being produced is challenging [26].

> **Key Points** (i) MU does not help with TDM opt-out violations; the damage is already done. (ii) MU may, however, help with downstream copyright violations in outputs. (iii) To avoid malicious unlearning, TDM opt-outs will have to be verified.

> **Open Problems** (i) Difficulty in identifying copyright-infringing works in a dataset. (ii) Difficulty of verifying whether model output owes to copyrighted data or generalization. (ii) Localizing and measuring memorization of copyrighted data is itself an open problem.

## 5 Discussion

MU might offer a path towards compliance with some AIA requirements, but it is not a silver bullet. Throughout this work, we have balanced enthusiasm for MU's capabilities with a clear-eyed view of its limitations. A recurring challenge across use cases — such as accuracy, bias, and confidentiality — is the difficulty of identifying and isolating harmful or low-quality data. In modern AI models, such information is often encoded in distributed representations, making precise removal difficult and risking forgetting useful knowledge.

In many cases, the target of unlearning (e.g., a fact or concept) lacks a discrete representation. Still, recent work in generative models shows promise: concept editing in diffusion models [63], data attribution [116], and inversion-based techniques [45] all offer ways to trace and remove implicit or emergent representations.

While some applications — like correcting mislabeled data to improve accuracy [73] — are feasible with today's methods, others (e.g., bias mitigation or copyright control) face steeper barriers. In some cases, MU may be an unnecessarily complex solution relative to alternatives (discussed further in Sec. 6). However, overlaps between applications (e.g., boosting both fairness and accuracy) suggest that well-designed MU interventions could serve multiple regulatory goals simultaneously.

One challenge that stands out is verification or auditability [111]. Throughout the paper we stress that auditability, i.e., the ability of parties, including regulators, to inspect and verify the efficacy of unlearning, is central to making MU viable for these AI regulation compliance use cases. However, as discussed, some of today's approximate MU methods offer only limited guarantees, complicating auditing. Instead, they rely on empirical proxies (e.g., attack success rates, performance recovery, or distributional similarity) rather than formal proofs. Consequently, these approaches lack strong guarantees that forgetting has been achieved or that residual model behavior no longer depends on the forgotten data. Until addressed, this absence of verifiable guarantees undermines both the auditability and the regulatory utility of MU. Going forward, we advocate for the development of formal forgetting guarantees that can underpin regulator-endorsed standards.

## 6 Alternative Views

The arguments against using MU as a tool for compliance with the AIA or other AI regulation would likely point to its shortcomings, trade-offs, and risks as well as the viable substitutes for MU in these scenarios. Some recent works, for example, broadly question whether MU can really achieve its goals, especially in the generative domain [26, 5, 137, 109]. Other works scrutinize MU's trade-offs around performance, privacy, security, and cost [125, 14, 133, 72, 137, 43]. These factors could reasonably make alternative methods more appealing for the AI regulation use cases highlighted in this position paper [138, 26]. For example, here are alternatives for each AI regulation use case discussed this paper and their possible advantages as compared to MU:

- **Accuracy**: Full retraining or improving data pipelines may be simpler and more reliable for correcting stale or mislabeled data, while MU may serve best as a fallback when those options are impractical [76, 73].

- **Fairness and bias mitigation**: Pre-, in-, and post-processing bias interventions may offer more holistic and verifiable control than MU, which may play a narrower, subtractive role [95, 51].
- **Confidentiality and privacy**: DP and access control may provide stronger, proactive safeguards against leakage, with MU offering only reactive, case-specific data removal after deployment [67, 50].
- **Security and data poisoning**: Robust or certified training and data-sanitization methods may deliver a more proactive defense, while MU may only serve as a reactive remediation step for excising identified poisoned data or backdoors [22, 97, 39].
- **GenAI risk reduction**: Alignment methods such as Reinforcement Learning from Human Feedback (RLHF) or guardrailing may more effectively constrain model behavior, with MU contributing mainly to remove residual unsafe or sensitive representations [5, 88, 118].
- **Copyright and intellectual property**: Dataset governance, licensing verification, fine-tuning, and output filtering may offer preventive compliance, whereas MU may better function as a post-hoc corrective tool for erasing memorized or stylistically infringing content [134, 132, 26].

These alternative approaches come with their own limitations. For instance, while some may consider DP [67] as a strong alternative to MU, several caveats deserve attention. First, DP mechanisms often struggle to balance tight privacy guarantees with acceptable model utility [106]. This trade-off becomes especially pronounced in high-utility applications. Second, unlike traditional privacy settings where protection is applied uniformly across all data points, MU typically targets a specific subset of data—the so-called "forget set." In large-scale training corpora that combine individually identifiable data with more publicly available content, applying DP globally may offer overly broad protections that are both inefficient and unnecessary [50]. Third, there are use cases where DP is not sufficient or optimal. For instance, if the objective is to remove a harmful or undesired behavior from a generative model (e.g., misinformation, bias, or offensive content), a DP-trained model may still require explicit MU interventions to mitigate such behaviors.

# 7 Conclusion

There are still sizable challenges that must be cleared before MU will be a viable tool for assisting compliance with the AIA (and, by extension, other AI regulations, which tend to feature recurring principles [40, 32]). To realize MU's potential for these AI regulation use cases, AI researchers should help solve the open technical problems logged by this position paper. Among other things, this includes work on identifying forget set data points, on resolving the privacy and performance trade-offs of MU, and on resolving the particular challenges posed by generative model outputs. Working collaboratively, we can all help unlock MU's potential to assist compliance with AI regulation and, by extension, help safeguard the important social values these regulations encode.

# 8 Impact Statement

In essence, this position paper suggests research directions that will help MU evolve into a better tool for assisting AI regulation compliance. Because these AI regulations tend to encode important ethical and societal values around health and safety, non-discrimination, and more, we believe the impact of this paper, in striving to advance AI regulation compliance through MU, will ultimately be the advancement of those important values as well.

## Acknowledgements

*Funding.* We wish to acknowledge the Google Academic Research Awards for their support.

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
