# OpenReview forum: "Position: Bridge the Gaps between Machine Unlearning and AI Regulation"
_NeurIPS.cc/2025/Position_Paper_Track — NeurIPS 2025 Position Paper Track Oral_

### Official Review · Reviewer_3mER · 2025-08-04

**Significance:** 1
**Presentation:** 2
**Rating:** 4
**Confidence:** 4

**Summary:**

This position paper argues that machine unlearning (MU) could assist compliance with AI regulations like the EU's Artificial Intelligence Act (AIA), but only if researchers bridge significant technical gaps between current capabilities and regulatory requirements. The authors catalog six potential AIA compliance applications for MU: improving accuracy, mitigating bias, defending against confidentiality attacks, addressing data poisoning, managing generative AI risks, and supporting copyright compliance. For each use case, they identify substantial limitations in current MU techniques, including difficulties identifying problematic data, lack of formal guarantees, and trade-offs between utility and forgetting quality. The paper concludes with a call for the AI research community to address these technical challenges to realize MU's regulatory potential

**Strengths:**

The paper provides the most comprehensive analysis to date of machine unlearning's potential regulatory applications. The systematic mapping of AIA requirements to technical capabilities is valuable for both researchers and policymakers. The honest assessment of current limitations and clear identification of research gaps could effectively guide future work. The technical analysis is rigorous while remaining accessible to non-experts

**Weaknesses:**

The analysis may overstate MU's potential benefits while understating the challenges of defining and measuring successful "unlearning" in regulatory contexts. Limited discussion of how regulatory compliance would be verified or audited in practice. Some use cases (particularly copyright) may be better addressed through alternative approaches that receive insufficient consideration. The paper doesn't adequately address whether current MU limitations reflect fundamental theoretical constraints versus engineering challenges. Also and foremost: the EU AI Act is a product safety regulation under the EU New Legislative Framework. It is thus 'techniques-agnostic' by default and this study might be more relevant for other regulations such as the EU GDPR, Data governance act, Data act etc

**Questions:**

How would regulators verify or audit successful machine unlearning in practice, given the current limitations in evaluation metrics? What evidence suggests that the identified technical gaps are solvable rather than fundamental limitations of the unlearning approach? How might the regulatory landscape's rapid evolution affect the relevance of these specific technical research priorities? Could you elaborate on the trade-offs between investing in MU research versus alternative approaches like differential privacy or architectural solutions?

**Alternative Position:**

Yes, and alternative positions are well-considered and named but not addressed

**Author Identification:**

No.

**Context:**

2

**Discussion:**

3

**Ethics:**

["NO or VERY MINOR ethics concerns only"]

**Position:**

No, the paper argues that a specific technical approach is superior to other approaches.

**Support:**

2

**Thoroughness:**

4

---

### Official Review · Reviewer_Jfy6 · 2025-08-05

**Significance:** 3
**Presentation:** 3
**Rating:** 5
**Confidence:** 4

**Summary:**

This paper examines the gap between the EU's AI Act (AIA) requirements and current machine unlearning capabilities. The authors explore potential applications of machine unlearning for AIA compliance across several domains: improving model accuracy, mitigating bias, defending against confidentiality attacks and data poisoning, addressing generative AI risks, and resolving copyright issues. The paper provides a comprehensive summary of existing machine unlearning approaches' key limitations and identifies critical open problems that require further research.

**Strengths:**

- The paper addresses a timely and important intersection between machine unlearning research and EU AI regulation, identifying critical technical gaps that must be bridged for practical compliance.
- The authors provide an effective literature review that clearly synthesizes existing machine unlearning approaches and systematically identifies their key deficiencies and limitations.
- The paper is well-written, clearly structured, and accessible to readers across different technical backgrounds.

**Weaknesses:**

(1) The paper's title references "AI regulation" broadly, but the analysis focuses exclusively on the EU's AI Act. While this represents a minor overclaim in scope, it could be easily addressed by adjusting the title to reflect the specific focus on EU regulation. A comprehensive analysis of the AIA alone remains valuable and substantive.

(2) The authors propose six distinct applications of machine unlearning for regulatory compliance, which provides a useful organizational framework. However, several of these applications exhibit significant conceptual and technical overlap that warrants deeper analysis. For instance, mitigating bias, controlling generative outputs, and addressing copyright issues all fundamentally relate to neural network memorization and may face similar algorithmic challenges. The paper would benefit from explicitly acknowledging these interconnections and discussing how they affect the design and evaluation of machine unlearning approaches. Additionally, since many existing machine unlearning algorithms claim to address multiple problems simultaneously, a more detailed algorithmic analysis examining which specific techniques are most suitable for each application domain would strengthen the contribution.

**Questions:**

Following the previous section:

(3) The paper falls a bit short of fully exploring the critical interaction between policy requirements and technical capabilities within the EU regulatory framework. Rather than offering a unique perspective on how machine unlearning should evolve to align with AI governance needs, the work reads primarily as a literature survey cataloging existing technical limitations. The paper would be significantly strengthened by providing concrete insights into how machine unlearning can be positioned as an effective regulatory compliance tool, including specific recommendations for bridging the gap between current technical capabilities and policy requirements. I encourage the authors to articulate a clearer vision for the co-evolution of machine unlearning research and AI regulation, addressing both the technical research priorities and the policy considerations necessary to make machine unlearning a viable compliance mechanism.

Question:
- When bias is exhibited in model behavior, are there any existing techniques that can attribute this to specific training data and then perform machine unlearning accordingly?

**Alternative Position:**

No

**Author Identification:**

No.

**Context:**

3

**Discussion:**

3

**Ethics:**

["NO or VERY MINOR ethics concerns only"]

**Position:**

Yes, the paper argues for or against a position related to machine learning.

**Support:**

3

**Thoroughness:**

4

---

### Official Review · Reviewer_648Q · 2025-08-09

**Significance:** 4
**Presentation:** 4
**Rating:** 8
**Confidence:** 5

**Summary:**

This position paper examines the gap between the current state of machine unlearning and its potential role in helping organizations comply with emerging AI regulations, using the EU’s Artificial Intelligence Act (AIA) as a case study. It catalogs possible applications of machine unlearning to AIA compliance, identifies technical shortcomings, and calls for research to address these gaps.

**Strengths:**

the paper is very well written.
it show case the bullet points  in GDPR and the relation with machine unlearning. the paper properly discusses intersection between the provided regulations and machine unlearning.

for each point, it first discusses how that rule is interpreted, or employed. then it shows the relation with unlearning and it then discusses the potential and downside of machine unleanring methods for that particular topic.


the paper is also very well positioned on the existing literature, regulations.

**Weaknesses:**

line 79> Hyperparameters .... > Please add reference for this statement.

line 112> Here risks ... > This statement can broadly cover many topics. It would be better specify the objective related to regulation for unlearning

**Questions:**

1- It would be interesting if the authors a direction for the research direction of the appointed topics for example in section "MU for AIA compliance: a catalog > Accuracy" authors very well described the what AIA expects by  "Accuracy: Improve accuracy per EU [35, Arts. 9, 15];", they also discussed the SOTA unlearning methods for that topic, and the advantages and disadvantages of those method. I think, the only missing point, is the proposed future direction for the unlearning research society to address the discussed concerns.

**Alternative Position:**

Yes, and alternative positions are trivial straw-man arguments

**Author Identification:**

No.

**Context:**

4

**Discussion:**

3

**Ethics:**

["NO or VERY MINOR ethics concerns only"]

**Position:**

Yes, the paper argues for or against a position related to machine learning.

**Support:**

4

**Thoroughness:**

5

---

### Note · Authors · 2025-09-03

**1-11 Submit Again:**

Unsure

**1-1 Submission Process:**

3

**1-2 Next Year:**

A proper rebuttal period, as in Neurip’s main track

**1-5 Thoughtful:**

5

**1-6 Supportive:**

5

**1-7 Technical Aspects Versus Position:**

3

**1-8 Gate Keeping:**

3

**1-9 Camera Ready Changes:**

We will (i) refine the title to explicitly emphasize the EU AI Act as our primary case study, (ii) incorporate additional references on hyperparameter trade-offs in unlearning methods, (iii) expand our open problems section with more concrete and domain-specific research directions (e.g., bias attribution, copyright attribution, generative risk), and (iv) strengthen our discussion of overlaps and complementarities across use cases (e.g., bias, generative risks, copyright) to clarify both their legal distinctiveness under the AIA and their shared technical mechanisms. We will also make our policy–technology recommendations more explicit by proposing pathways for regulators and researchers to co-develop auditable standards.

**3-1 Review Response1:**

648Q

**3-2 Reaction To Review1:**

We thank the reviewer for their thoughtful and positive assessment of our work, including recognition of its clarity, rigorous positioning in the literature, and systematic treatment of how machine unlearning intersects with the EU’s AI Act (AIA). We address their comments below.

**On hyperparameter trade-offs (Line 79)**

We now cite prior works showing MU trade-offs’ sensitivity to hyperparameters and regularization (e.g., Bourtoule et al. [2021], Graves et al. [2021], Liu et al. [2024]).

**On clarifying “risks” (Line 112)**

We revised this passage to tie risks directly to AIA objectives: (i) health and safety, (ii) fundamental rights (e.g., non-discrimination), and (iii) malicious interference (e.g., confidentiality or poisoning attacks).

**On future research directions**

In the revised version, we have added a new subsection at the end of each catalog entry outlining specific open problems and directions. For example:

-- For Accuracy, we propose new methods for reliably identifying mislabeled and out-of-date samples, and benchmarks for measuring whether MU reliably improves accuracy when removing mislabeled data, rather than assuming retraining equivalence.

-- For Bias, we call for counterfactual-based methods to identify features that drive group disparities and for unlearning approaches that preserve subgroup accuracy.

-- For Confidentiality, we emphasize the need for MU approaches that can withstand emergent adaptive attacks, complemented by formal guarantees.

-- For Data Poisoning, we suggest hybrid MU + forensic methods that first detect contaminated subsets before applying selective forgetting.

-- For Generative Outputs, we advocate for scalable “concept-level” MU that can target latent harmful knowledge without broadly degrading utility.

-- For Copyright, we highlight the open challenge of attributing memorized versus generalized content, and propose MU methods combined with watermarking for compliance verification.

**3-3 Review Response2:**

Jfy6

**3-4 Reaction To Review2:**

We thank the reviewer for their careful reading and constructive feedback.

**Scope of title**

We agree that the title could suggest a broader scope. Our intent was to use the AIA as a representative case study, given it is the most advanced and influential regulatory framework. Many of its principles recur in other jurisdictions (e.g., U.S., Brazil, Canada, Australia). Thus, our focus is strategic rather than narrow. Still, we are happy to refine the title to make the AIA focus explicit while retaining broader relevance.

**Overlap across applications**

We acknowledge overlaps between bias, generative outputs, and copyright, which often share mechanisms like memorization. Our modular treatment reflects the fact that these are distinct obligations in the AIA (Articles 9, 15, 53, 55). Regulators treat them separately, even if similar technical remedies apply. We also note overlaps in the Discussion and will strengthen this by drawing out common algorithmic techniques.

**Policy–technology interaction and actionable recommendations**

We disagree that our work reads primarily as a survey. A key contribution is our catalog aligning each AIA requirement with MU research gaps (Sections 4.1–4.6). We identify actionable open problems such as bias counterfactuals, memorization localization, and formal forgetting guarantees, and call for regulator-endorsed standards as a roadmap for co-evolution of research and governance. We can further emphasize this by recommending: (i) regulators provide clearer technical specifications for “forgetting,” and (ii) researchers prioritize scalable verification and attribution methods.

**Bias attribution and unlearning**

Recent works support this direction. Data attribution methods can localize bias contributions, enabling MU at the feature or sample level. While still limited in scalability and precision, our paper highlights the need to formalize such pipelines as a core MU–regulation research agenda.

**3-5 Review Response3:**

3mER

**3-6 Reaction To Review3:**

We thank the reviewer for the careful reading and constructive comments. We would like to clarify points where our contribution may have been underestimated.

**On overstating MU’s potential benefits**

We explicitly balance enthusiasm with caution throughout. Each of the six use cases (accuracy, bias, confidentiality, poisoning, generative risks, copyright) includes explicit limitations and open problems (Sec. 4). For example, MU cannot remedy data deficits in bias or undo TDM opt-out violations in copyright. Our novelty lies in systematically mapping both opportunities and limits to guide research.

**On verification and auditing**

We agree this is critical and highlight it as a core call to action (Sec. 5). Our contribution is to link verification directly to regulatory needs, stressing that auditable guarantees are essential for compliance, not just academic interest.

**On solvability of technical gaps**

We distinguish between engineering challenges (e.g., scalable forget-set identification) and deeper conceptual hurdles (e.g., unlearning latent knowledge in generative models). By organizing these across six compliance applications, we provide a roadmap that separates promising avenues from fundamental barriers.

**On alternative approaches**

We acknowledge MU is not a silver bullet and compare it with differential privacy and post-training alignment (Sec. 7). MU is complementary: it enables selective removal, while DP offers global guarantees with higher utility trade-offs. Understanding such synergies is crucial.

**On regulatory context**

The AIA is technique-agnostic, which is why our analysis matters: providers may use MU to demonstrate compliance. The AIA also imposes obligations beyond GDPR (accuracy, bias mitigation, poisoning defense, copyright), making it a timely and distinct case. As noted in our conclusion, our principles generalize to GDPR, the Data Governance Act, and the Data Act.

---

### Meta-Review · Area_Chair_echt · 2025-09-12

**Rating:** 6
**Confidence:** 3

**Strengths:**

Reviewers stated the following strengths:
1. Timely contribution to the field of AI regulation. Discussed the important intersection of machine unlearning and EU AI regulation, identifying key technical gaps for compliance.
2. Offered comprehensive analysis: Provides a detailed mapping of GDPR/AIA requirements to unlearning methods, offering thorough treatment of the regulatory-technical intersection to date. Effectively synthesizes existing unlearning approaches, situates them in current regulatory contexts, and highlights their limitations.
3. All reviewers mentioned that the paper is clear structured and with good readability. For each regulatory point, the paper interprets the rule, relates it to unlearning, and discusses both potential and downsides of existing methods.

**Weaknesses:**

Reviewers pointed out the following suggestions for the authors to improve their work:

1. Elaborate overlap among proposed applications. The paper would benefit from explicitly acknowledging these interconnections and analyzing how they shape the design and evaluation of unlearning methods.

2. Verification and auditability require deeper discussion.

3. Further extension of alternative views. Some use cases (e.g., copyright) might be better addressed with other approaches, but these alternatives are only briefly mentioned. A comparative discussion would provide better context.

There are also other weakness points raised by reviewers, but I agree with the authors that those points might not be valid.

**Questions:**

There is no further question from me. Authors have answered reviewers' questions through author survey.

**Ethics:**

There is no ethical violations or concerns raised by reviewers

**Thoroughness:**

2

---

### Decision · Program_Chairs · 2025-09-26

Accept (Oral)